# Lake ecosystem tipping points and climate feedbacks

Dag O. Hessen[1], Tom Andersen[1], David Armstrong McKay[2,3], Sarian Kosten[4], Mariana Meerhoff[5,6], Amy Pickard[7], Bryan M. Spears[7]

[1] Department of Biosciences, Centre for Biogeochemistry in the Anthropocene, University of Oslo, Box 1066 Blindern, 0316 Oslo, Norway

[2] Global Systems Institute, University of Exeter, North Park Road, Exeter, EX4 4QE, UK

[3] Stockholm Resilience Centre, Stockholm University, Stockholm, Sweden

[4] Department of Aquatic Ecology and Environmental Biology, Radboud Institute for Biological and Environmental Sciences, Radboud University, Nijmegen, the Netherlands

[5] Department of Ecology and Environmental Management, Centro Universitario Regional del Este (CURE), Universidad de la República, Cachimba del Rey s/n Maldonado, Uruguay.

[6] Department of Ecosciences, Aarhus University, Aarhus, Denmark.

[7] UK Centre for Ecology & Hydrology (Edinburgh), Bush Estate, Penicuik, Midlothian, EH26 0QB, UK

*Correspondence to*: Dag O. Hessen (d.o.hessen@mn.uio.no)

**Abstract**
Lakes and ponds experience anthropogenically-forced changes that may be non-linear and
sometimes initiate ecosystem feedbacks leading to tipping points beyond which impacts become
hard to reverse. In many cases climate change is a key driver, sometimes in concert with other
stressors. Lakes are also important players in the global climate by ventilating a large share of
terrestrial carbon back to the atmosphere as greenhouse gases, and will likely provide substantial
feedbacks to climate change. In this paper we address various major changes in lake ecosystems,
and discuss if tipping points can be identified, predicted, or prevented, as well as the drivers and
feedbacks associated to climate change. We focus on potential large-scale effects with regional
or widespread impacts, such as eutrophication-driven anoxia and internal phosphorus-loading,
increased loading of organic matter from terrestrial to lake ecosystems (lake "browning"), lake
formation or disappearance in response to cryosphere shifts or changes in precipitation to
evaporation ratios, switching from nitrogen to phosphorus limitation, salinization, and the spread
of invasive species where threshold-type shifts occur. We identify systems and drivers that could
lead to self-sustaining feedbacks, abrupt changes and some degree of resilience, as opposed to
binary states not subject to self-propelling changes or resilience. Changes driven by warming,
browning, and eutrophication can cause increased lake stratification, heterotrophy (browning),
and phytoplankton or macrophyte mass (eutrophication), which separately or collectively drive
benthic oxygen depletion, internal phosphorus-loading and in turn increase greenhouse gas
(GHG) emissions. Several of these processes can feature potential tipping point-thresholds,
which further warming will likely make easier to surpass. We argue that the full importance of
the vulnerability of lakes to climate and other anthropogenic impacts, as well as their feedback to
climate is not yet fully acknowledged, so there is a need both for science and communication in
this regard.

## 1. Introduction

In natural sciences, the hysteretic behaviour of lakes (Scheffer et al. 2007) has informed the concept of tipping points at the ecosystem level, following the development of the alternative stable states theory in shallow lakes (Scheffer et al. 1993). Given the global vulnerability of freshwaters and the pervasive nature of major pressures acting upon them (e.g. nutrient pollution, over-extraction, and climate change), tipping points in these systems could have significant societal impacts, including on human and environmental health, clean water and food production, and climate regulation. The capacity to detect discontinuous ecosystem responses to pressure changes in natural systems has been challenged (e.g. Hillebrand et al. 2020; Davidson et al. 2023). Nevertheless, there are several studies that have reported the occurrence of tipping points even if they are difficult to detect (Lade et al, 2021; Seekell et al. 2022), such as shifts from one alternative state to another in small shallow lakes, the most common lake type globally (Messager et al., 2016).

Some types of changes can be classified as *binary*, i.e. either-or situations at the system level. Increased temperature and/or reduced precipitation may induce negative water balance and shrinking of water volumes to the level where lakes or ponds simply disappear. Many lakes worldwide are facing reduced water volumes, but perhaps most striking is the widespread loss of high-latitude waterbodies, from Arctic or sub-Arctic ponds to wetlands or bogs. Such phenomena may qualify as one type of tipping point, but are not self-propelled by internal feedbacks *per se*, but rather by higher evaporation to precipitation ratios (Smol and Douglas 2007) or permafrost thaw (Smith et al. 2005; Webb et al. 2022; Smol 2023). While most tundra-ponds and other small waterbodies hardly qualify as *lakes* (Richardson et al. 2022), we mostly use the word lake through the text for simplicity, yet it will be evident from the context of wording where we specifically refer to ponds.

The question of what constitutes a "sudden" system shift, alternative stable states and hysteresis depends too on what is considered a relevant time span; days, years, decades or centuries. Also, systems may have alternative states that are not necessarily fixed over long time-spans, hence the phrase "stable" should be used with caution, so to the term 'hysteresis' infers little on the strength of the regulating processes. Uncertainty also remains on the geographical extent of tipping points in lakes and the wider relevance for the Earth's climate system. Single lakes or local areas may experience non-linear or abrupt changes caused by local drivers, but we

here focus on potential tipping points of global or regional relevance, and with relevance to the
climate system.

Empirical analyses, process modelling and experimental studies are advanced for shallow

lakes, providing a good understanding of lake ecosystem behaviour around tipping points. There
are related concepts in the literature (regime shifts, catastrophic shifts, forward switches, etc.),
and there clearly many aspects of abrupt changes in nature and society that could be labelled a
tipping point (Carrier-Belleau et al. 2022), but here we adopt the definition of a tipping point
occurring when self-sustaining change in a system is triggered beyond a forcing threshold,
typically starting with positive feedback loops, then entering a runaway phase before finally the
tipping-point brings the system into a different alternative state (Nes et al. 2016; Lenton et al.
2023). For example, the well documented increase of phosphorus (P) loading across European
lakes in the last century (e.g. from agricultural and waste water pollution) has uncovered critical
loading thresholds beyond which lakes can shift rapidly from a clear water, submerged
macrophyte rich state to a turbid, phytoplankton dominated state (Scheffer et al., 2001; Jeppesen
et al., 2005; Tátrai et al. 2008), and vice versa, when nutrient loading decreases. One of the
theoretical implications is that to induce a switch back to the initial state the nutrient loading
should be reduced to a lower threshold before the shift might be possible (hysteresis). Adding to
such well-described and mechanistically well understood changes, there is a range of phenomena
that may be perceived as tipping points. Hence, to provide structure to this complexity a range of
tipping point candidates should be scrutinized against a common assessment approach. To
qualify as tipping points, phenomena should not just be isolated phenomena in single lakes, but
be more general and hold for specific (and widespread) types of lakes or waterbodies. Such
phenomena may thus in the future occur across geographically distinct lake populations
experiencing similar environmental change. In this way, the potential for identifying regional or
global scale changes can be framed (Fig. 1).

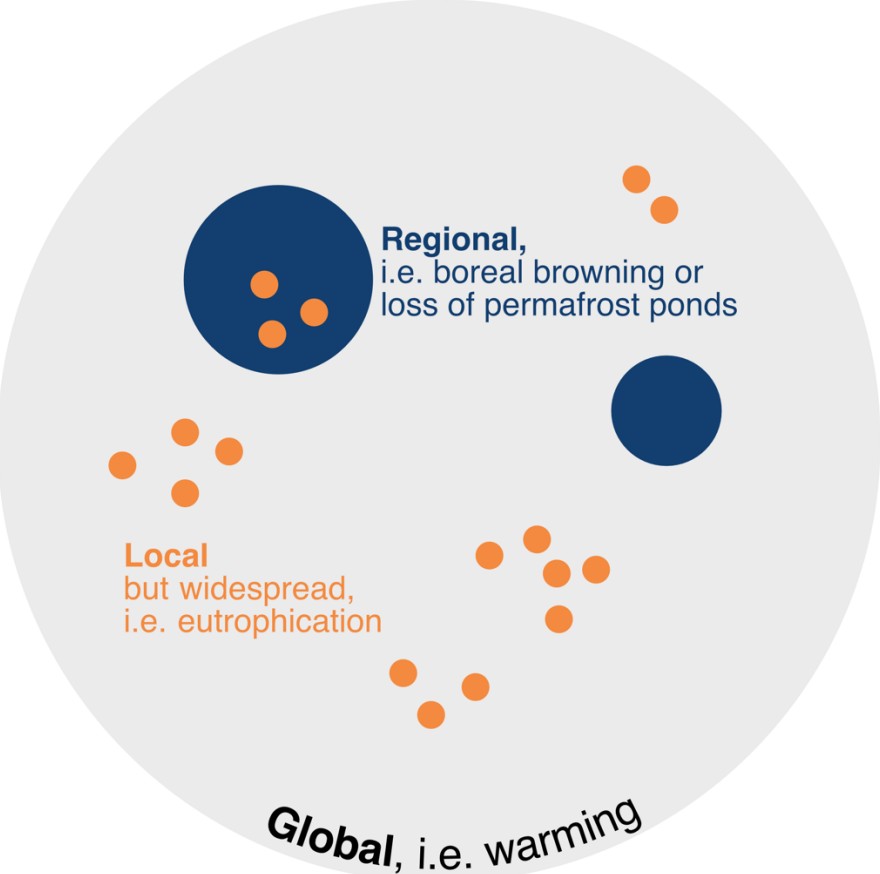


Fig. 1. Impacts at levels that may qualify for tipping points at relevant spatial scales. Regional or
biome-wise effects could be loss of ponds and lakes due to permafrost thaw and/or increased
loadings of DOM in the boreal biome or salinization. Also local, but widespread changes such as
anthropogenic eutrophication of lakes in populated or intense agricultural areas would have
large-scale impacts. Lakes worldwide shows a warming trend, hence a global impact.


It is well established that lakes are sensitive to the effects of climate change, including
warming and changes in precipitation and storminess (e.g., Adrian et al., 2009; Meerhoff et al.,
2022). Emerging evidence suggests that lakes and ponds may also play an important role in
climate regulation, through both the emission of greenhouse gases (i.e. $CO_2$ and $CH_4$,
predominantly $CH_4$, Downing et al., 2021) and carbon burial (Anderson et al., 2020). Lakes and
rivers are impacted by climate change and other anthropogenic pressures globally, but they also
provide strong feedbacks to the global climate systems and carbon (C) cycle (Cole et al. 2007;
Tranvik et al. 2009), despite comprising a small part of global water extent. While global
estimates of net greenhouse gas (GHGs) emissions from lakes remain poorly constrained, there is
general consensus that a significant fraction of terrestrially fixed C is degassed to the atmosphere
via surface waters. Cole et al. (2007) conservatively estimated that inland waters annually
receive some 1.9 Pg C y$^{-1}$ from the terrestrial landscape, of which at least 0.8 Pg C y$^{-1}$ is
returned to the atmosphere through water to atmosphere GHG exchange. Later estimates revised
this global GHG exchange term, to include evasion rates, at 2.1 Pg C yr$^{-1}$, from lakes, rivers and
reservoirs (Raymond et al. 2013). Notably, boreal lakes are important conduits of $CO_2$ release to
the atmosphere, estimated to be equivalent to the annual $CO_2$ release from forest fires, globally
(Hastie et al. 2017). Under a high $CO_2$-emission scenario and as a result of increased terrestrial
NPP, $CO_2$ emissions from boreal lakes are projected to increase by 107%, showing the coupling
between the terrestrial and aquatic C cycle (Hastie et al. 2017).

This significant role of surface waters for GHG-emissions is also highly relevant, but

poorly constrained, in both national and global C-budgets (Lindroth and Tranvik 2021). The
relationship between inputs of organic C and nutrients is a key determinant of the balance
between heterotrophic and autotrophic processes, determining the biodiversity, community
composition and food web structure, as well as the productivity-to-respiration (P:R) ratio. And
so, it is relevant to consider the extent to which potential tipping points may drive, or be driven
by, climate change, leading to higher level feedbacks to the Earth's climate system.

Here, we discuss candidate tipping points in freshwaters reported in the literature (based

on literature searches including the term 'tipping point' and either 'lake' or 'pond') as well as
experience and insights of the author team. The discussion on each is constrained to waterbody
categories with the potential for global or at least regional or biome-scale relevance. In this
context we also constrain the discussion to potential tipping points that are more generic, at least
carrying regional or biome-wide impact, and that could have feedbacks to the climate, while not
necessarily being driven or triggered by climate change *per se*.

We identify 6 candidate categories for tipping points at a relevant scale in this context

(regional to global impact), and for each of the categories we discuss whether observed changes
can be categorised as tipping points according to the definition above. We also address climatic
and other drivers and consequences, including potential feedbacks to the climate system, and
wider societal implications, with emphasis on the most relevant and influential categories.

## 2. Candidates and categories of lake tipping points

In principle, any abrupt or sudden stress imposed on a waterbody could result in specific impacts, i.e. toxic waste or toxic treatments (e.g. rotenone to kill off undesired species; runoff of herbicides inadvertently killing aquatic plants), hydrological alterations by impoundment or canals, and stocking or immigration of new (often exotic) species. In some cases, when stressors are removed the system will return abruptly to its original state. To qualify as a tipping point, here, we consider that system response should be self-sustaining and involve positive feedbacks, in line with the criteria set out in Nes et al. (2016) and Lenton et al. (2023). To be relevant in a wider context, the tipping point should be more generic to certain types of impact, certain types of waterbodies, and potentially also have feedbacks to the climate in terms of GHG-emissions. We have identified 6 stressors that may trigger a freshwater ecosystem to cross a tipping point (Table 1) and scrutinise them in turn below.

Table 1. Candidate events from the literature with potential to occur at local to regional scales, their association with climate change, and whether tipping points and hystereses have been associated with them. Brackets indicate higher uncertainty.

| Type of event | Local, common | Regional | Climate driver | Climate feedback | Tipping point | Hysteresis |
|---|---|---|---|---|---|---|
| Eutrophication driven water anoxia and internal P-loading | x | | x | x | x | x |
| Increased loadings of DOM | | x | x | x | (x) | (x) |
| Disappearance/ appearance of waterbodies | | x | x | x | x | (x) |
| Switch between N and P limitation | | x | x | (x) | | |
| Salinization | | x | x | x | | (x) |
| Spread of invasive species | x | (x) | (x) | | | (x) |

### 2.1. Eutrophication, anoxia and internal P-loading

Eutrophication is one of the most pervasive stressors on fresh water and coastal systems. Although it may naturally occur due to inputs from the watershed or from biota translocating nutrients from connecting ecosystems, eutrophication is a largely human-induced phenomenon. The main causes of cultural eutrophication have varied across time and regions. However, it is widely accepted that the main current cause of eutrophication is the change in land use in

watersheds, and, particularly agricultural activities driving diffuse nutrient pollution (as well as
other agrochemicals) (Moss 2008; Schulte-Uebbing et al., 2022). Agriculture, with myriad
impacts on fresh waters that go well beyond nutrient pollution (Moss, 2008), has been identified
as a major driver of ecosystem shifts and tipping points (Gordon et al., 2008).

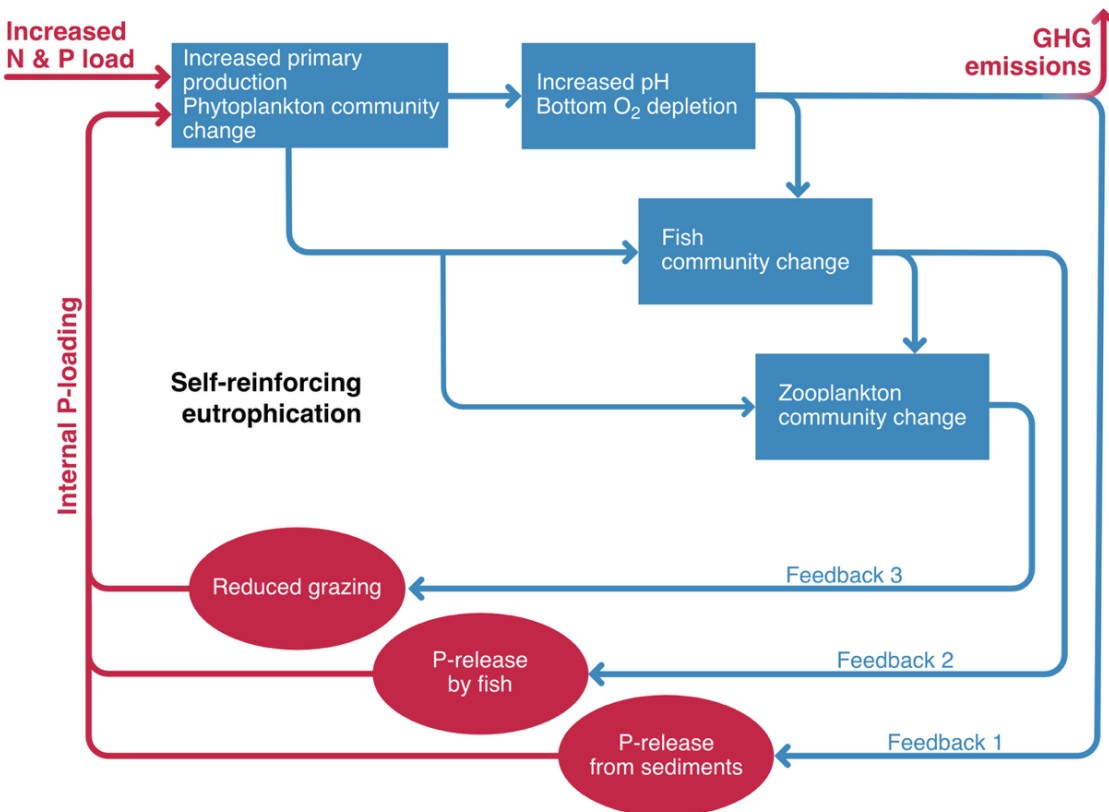


Fig. 2. Feedback loop diagram for eutrophication, demonstrating key feedbacks that can amplify
P-loading, and beyond a tipping point cause self-sustaining change. Eutrophication and internal
mobilization of P cause high algal biomass, decrease benthic oxygen or anoxia, and consequently
also increased greenhouse gas emissions. Blue denotes primary responses, red the secondary
feed-back responses as well as the key driver and consequence (nutrient loading and GHG-
release).

The mobilisation of P from lake bed sediments, a process known as internal loading
(Sondergaard et al., 2001), plays a key role in hysteresis in lakes following the reduction of P
loading from the catchment (Boström et al. 1982; Jeppesen et al. 1991; Spears & Steinman
2020). In this context, hysteresis can be strengthened by eutrophication-driven biological
changes in fish composition and size structure that have cascading effects on zooplankton and
phytoplankton as well as strong impacts on fish-mediated nutrient cycling (Brabrand et al. 1990).
This in turn will maintain a system with deep water anoxia and high nutrient load, supporting the
release of GHGs (Fig. 2).

*Feedbacks and Tipping points*
The phenomenon of eutrophication is local, but widespread, and likely to worsen in its
manifestations as a result of climate change (Moss et al., 2011; Meerhoff et al., 2022). In
particular, the process of internal loading may be enhanced by lake warming (Jeppesen et al.,
2009) due to an increased metabolism of bacteria and an acceleration of biochemical reactions.
Warming also increases stratification and the duration and strength of stratification, also
promoting anoxia (Maberly et al. 2020; Woolway et al. 2020; 2022). As a case example, this
phenomenon is well documented by the recent study of the Danish, shallow and highly eutrophic
lake Ormstrup (Davidson et al. 2024). Increases in precipitation, and high intensity rainfall
events, are also expected to significantly increase runoff of P from agricultural catchments to
surface freshwaters (Ockenden et al., 2017), further promoting eutrophication and its
manifestations.
The different states of shallow lakes can feedback differently on climate by either
reducing or increasing GHG emissions (Hilt et al. 2017). Clear and turbid lakes differ in their
$CO_2$ emissions due to the magnitude of $CO_2$ uptake by primary producer photosynthesis. Efflux
of $CO_2$ appears to decrease when submerged macrophytes establish after the reduction of
nutrient loading (Jeppesen et al., 2016). Submerged-macrophyte dominated shallow lakes tend to
emit lower $CH_4$ by ebullition and diffusion than phytoplankton dominated, turbid lakes (Colina
et al., 2022; Davidson et al. 2018). The turbid state in particular feeds back on climate since
warming and eutrophication-induced water anoxia could offset increased $CO_2$-fixation by
blooms or by macrophytes as lower oxygen levels stimulate methane ($CH_4$) emission, with $CH_4$
emissions from eutrophic systems expected to increase with 6-20% with each degree of warming
(Aben et al. 2017).
The eutrophication and warming-associated shift from submerged macrophyte dominance
to phytoplankton or floating plant dominance may also strongly increase greenhouse gas
emissions, particularly $CH_4$ (Aben et al. 2022). Cyanobacterial blooms, a typical manifestation of
eutrophication and high internal P-loading, can both promote $CO_2$ sequestration and produce
CH$_4$. CH$_4$ can be produced even under oxic conditions as a by-product of photosynthesis (Bižić
et al., 2020). Blooms often create anoxic layers in surface sediments or through the water column
after their collapse, favouring the production and transport of CH$_4$ via methanogenesis (Li et al.,
2021; Yan et al., 2017). Cyanobacterial blooms are thus considered a key mechanism by which
eutrophication has a positive feedback on climate change (Bižić 2021; Yan et al., 2017).
Although increased inputs on N from atmospheric deposition or catchment runoff are the main
causes of elevated N$_2$O release from lakes (Yang et al. 2015), warming also impacts aquatic N$_2$O
emissions. N$_2$O emissions are estimated to increase by 8 – 14% for each degree of warming
(Velthuis and Veraart 2022), highlighting another strong climate feedback.
Despite the fact that nutrient loading is still the major driver of eutrophication (including
algal blooms; e.g. Bonilla et al., 2023), climate change is also expected to promote
eutrophication (Moss et al., 2011; Meerhoff et al., 2022). Indeed, interaction between
temperature and trophy has been observed to produce synergistic emission responses in
experimental lakes (Davidson et al., 2018) and warming alters resident microbial communities to
favour methanogenesis over methanotrophy (Zhu et al., 2020). It is thus likely that warming
decreases the nutrient thresholds for a tipping point leading to a shift to an alternative state in
shallow lakes and ponds.
The predominantly amplifying influence of climate change on eutrophication-driven
tipping points in lakes provides a mechanism for coherent threshold exceedance beyond the local
scale, with more widespread eutrophication-induced tipping points expected with further
warming. However, despite the dearth of studies that generate bi-directional carbon flux data to
assess the balance between emission and burial in lakes, it remains unknown whether the climate
feedback can be buffered by projected eutrophication-driven increases in lake carbon burial
(Anderson et al. 2020). Moreover, robust projections are lacking for climate impacts on
eutrophication, with no emergent regional to global warming threshold identifiable beyond
which a nonlinear increase in these localised tipping points occur (Grasset et al. 2020). In
general, tipping points becomes harder to predict in a warmer climate (Kosten et al. 2009).
**2.2. Increased loadings of DOM in the boreal biome**
Over thousands to millions of years, the feedback between terrestrial vegetation and aquatic
productivity has been essential for the evolution of the atmosphere and the global climate
(Beerling 2007). Vegetation serves not only as a major C pool and eventually a source of total
organic carbon (TOC) in boreal areas, but it also promotes root exudates of $CO_2$ and organic C.
This enhances weathering rates thereby increasing the flux of nutrients (P, N, Si, Fe, Ca and
carbonate ($CO_3$)) (Humborg et al. 2004; Hessen et al. 2009) to surface waters. The availability of
nutrients subsequently enhances aquatic productivity, and thereby C-sequestration. In addition,
the carbonate species are important for buffering capacity towards acidification in fresh water
and marine systems. On different timescales there is thus a range of feedback mechanisms
between terrestrial and aquatic ecosystems that demands a better understanding. Tracking past
history (Holocene) tree-line, forest cover and lake sediments, revealed a strong and consistent
link between climate, forest cover and lake TOC (Rühland et al. 2003; Rosén 2005). Thus, at
least on the centennial scale, there is a strong temporal TOC-link between terrestrial and aquatic
systems. Allochthonous C derived either directly as leachate from litterfall and roots or indirectly
via partial decomposition of organic matter in the soils, constitutes the (by far) dominant pool of
dissolved organic matter (DOM) in boreal freshwaters, hence forest cover and fraction of bogs
and wetland areas in the catchment are key determinants for the concentration and color of this
terrestrially derived chromophoric DOM (Dillon and Molot 1997; Kortelainen et al. 2006;
Larsen et al. 2011a).

Since terrestrially derived C is a main determinant of freshwater C, any changes in

terrestrial primary production and export of organic C will invariably also increase aquatic
outputs of $CO_2$. Increased terrestrial productivity has been linked to "$CO_2$-fertilization" (Huang
et al. 2007) yet these $CO_2$ effects will be constrained by N-availability. Elevated N-deposition
due to human emissions has driven an increase in the forest C sink in tandem with the $CO_2$-
fertilization effect, while at the same time also increased the deficiency of P (and other key
elements allocated to tree biomass) (De Vries et. al. 2006).

Increased export of terrestrially derived DOM to lakes and rivers in boreal regions

("browning") is a widespread phenomenon partly linked to reduced acidification, but also driven
by land-use changes (notably afforestation) and climate change ($CO_2$-fertilization of forests,
warming and hydrology) (de Wit et al., 2016; Creed et al. 2018; Monteith et al., 2023). An
empirically based space-for-time model of changes in the Normalised Difference Vegetation
Index (NDVI) under a $2^\circ$ C climate scenario predicts continued browning of boreal lakes (Larsen
et al. 2011b). Forest dynamics are slow, however, hence space-for-time scenarios projecting
increased flux of TOC from catchments owing to increased forest cover could require centuries
to play out. Thus, catchment properties governing *production* of TOC such as forest size and
fraction of bog and wetland areas could very well be temporally decoupled from the export,
especially considering the large stock of organic matter typically present in boreal catchments.

Time series analysis (30 years) of data from 70 Norwegian catchments and lakes

provided evidence for a tight temporal coupling between the decadal increase in land "greening"
(with NDVI as a proxy) and lake browning (with TOC as a proxy) (Finstad et al. 2016), and the
browning on northern lakes can, to a large extent, be attributed to recent afforestation (Kritzberg
2017; Skerlep et al. 2020). The prominent "greening" by increased vegetation cover trend in
many boreal and alpine regions (Guay et al. 2014) and increase in forest volume (cf. Opdahl et
al. 2023) will thus have bearings on lakes and rivers in these regions. There are a number of
confounding explanatory drivers for this greening: warming, elevated $CO_2$, accumulated nitrogen
deposition and changes in grazing activities as well as forestry practices are all implicated. An
extended growing season has also been recorded (Barichivich et al. 2013), and elevated levels of
$CO_2$ per se may contribute to this (Piao et al. 2006). Collectively, these changes in environmental
drivers and pressures yield an increase in terrestrial net primary production (NPP), notably at
high latitudes (Forkel et al. 2016). Since a significant fraction of the terrestrial NPP will be
exported to surface waters as DOM, it means that terrestrial greening could lead to freshwater
browning.

The role of forest cover is further accentuated by a need for a carbon-negative future (i.e.

net drawdown of $CO_2$ from the atmosphere) where widespread afforestation is the only currently
feasible means of reducing atmospheric concentrations of $CO_2$ beyond the continued action of
natural carbon sinks (MacDougall et al., 2020). However, such afforestation also comes with
climate costs, both in terms of decreased albedo (Betts and Ball 1997; Bathiany et al. 2010;
Lawrence et al. 2022) and as argued above, the potential for increased production and degassing
of GHGs from surface waters. Enhanced primary production in forested catchments stimulated
by reactive N deposition has, by increasing the pool of C available for fluvial export, been linked
to increased C burial in northern lakes over the past two centuries (Heathcote et al. 2015). Again,
this highlights the need for improved understanding of the balance between C emissions and
burial in lakes in response to browning (Williamson et al., 2015) and other identified stressors in
order to better constrain climate feedbacks. Browning will also promote darkening of coastal
waters with as yet unknown climate feedbacks (Opdal et al. 2023). The question that remains is
whether these terrestrial and aquatic responses are directly coupled in time, or if there is a
delayed aquatic response in the order of decades or even millennia. Another question is how the
$CO_2$ in itself could boost these processes, and how this skewed C-supply to autotrophs could
affect land-aquatic interactions?

Wide-scale shifts in boreal lakes caused by increased loadings of DOM can promote a

prolonged and more intensified stratification period (implications summarized above, described
for DOM by Spears et al., 2017), amplified by warming. Increased terrestrial DOM loadings
intensify net heterotrophy in the systems (i.e. through increased light attenuation and increased
access to organic C for heterotrophic bacteria) (Hessen et al. 1990; Karlsson et al. 2007; Thrane
et al 2014; Horppila et al. 2023). A well explored case study as an example of these impacts,
which is also linked directly to the tipping point concept, is the Swedish, boreal brownwater lake
Härsvatten where a long-term study clearly links loadings of DOC to anoxia (Spears et al. 2017).
While at present the thresholds around these effects have not been well constrained, the impacts
may be significant at the global scale for GHG emissions (Tranvik et al. 2009) and regionally for
coastal primary producers (Opdal et al. 2019). Given the strong empirical links between drivers
and consequences, it means that impacts and feedbacks can be predicted qualitatively, while not
yet quantitatively.

The temporal aspect also deserves further attention. If the main source of browning is

afforestation, responses will proceed slowly compared with cases where reduced acid deposition
is the main driver, yet both drivers operate on decadal timescales. In the latter case, the browning
could represent a *re-browning* (Meyer-Jacob et al. 2020).

*Feedbacks and tipping points*
The links and feedbacks between climate to land to lakes and back to climate in terms of
increased GHG-emissions is conceptually well understood, and also the main drivers for the
specific GHGs ($CO_2$, $CH_4$ and $N_2O$) in boreal areas are understood (Yang et al. 2015; Wik et al.
2016; Valiente 2022). However, the question of whether these feedbacks can result in tipping
points by becoming self-sustaining beyond a threshold is not yet settled. Most boreal lakes are
net heterotrophic and thus conduits of $CO_2$, often also $CH_4$, due to high concentrations of DOM
and common deep-water anoxia. A shift from net autotrophy to net heterotrophy would classify
as a binary shift, yet with a strong, positive climate feedback. If it eventually leads to oxygen
depletion and cascading feedbacks then it would qualify as a tipping point. However, there
would be a time delay between the two events, with the latter being the critical tipping event.
There is also a commonly reported unimodal response in lakes to increased loadings of DOM,
typically around 5 mg DOC $l^{-1}$ (Karlsson et al. 2007; Thrane et al 2014), where increases in
DOM below the threshold may promote NPP and thus $CO_2$ drawdown due to N and P associated
with DOM, while reduced NPP and increased degassing of $CO_2$ (and $CH_4$) will take place above.
We thus propose two types of large-scale potential tipping points, one related to anoxia, the other
to DOM-concentrations, yet both are related to increasing load of terrestrially derived DOM
across the boreal region.

**2.3. Disappearance/appearance of waterbodies**
A global reduction in lake water storage (Yao et al., 2023), and the climate-driven creation or
disappearance of water bodies is a crucial issue. Loss of water-bodies due to overuse, warming
or drought pose a major threat to vulnerable, freshwater resources, also by deteriorating water
quality or salinization (cf. below). The most dramatic warming has already taken place in the
high Arctic with temperature increases up to 3 °C over the past few decades (Wang et al. 2022),
further promoting the onset of permafrost thaw (Langer et al. 2016). Current and future
permafrost thaw and glacier melting can both create new waterbodies and drain old, providing a
strong link to the fate of the cryosphere (Smith et al. 2005; Olefeldt et al. 2021). Such small, but
numerous waterbodies residing on permafrost over large geographical scales in Eurasia and
North-America are currently among the most vulnerable water-bodies globally (Smol and
Douglas 2007; Heino et al. 2020). They host species-poor but specific communities of
invertebrates (Rautio et al. 2011; Walseng et al. 2021) of vital importance for birdlife and other
biota. Warming may also affect these waterbodies indirectly via glacier melt, increased inputs of
organic C, fertilisation by increasing populations of geese (caused by climate change), and
consequently changes in microbial communities and increased GHG emissions (Eiler et al.
2023). Thus, by their sheer number these systems may also serve as increasingly important
conduits of GHGs and historical soil carbon stocks to the atmosphere (Laurion et al. 2010;
Negandhi et al. 2016) and play an important role in mediating nutrient delivery to the polar
oceans (Emmerton et al., 2008), potentially affecting global NPP (Terhaar et al., 2021). While
the main problem is loss of water bodies affected by warming-induced increased
evapotranspiration rates (Smol and Douglas 2007) and permafrost thaw (Smith et al. 2005), there
are also cases where collapsing palsas and thermokarst areas create new waterbodies, and these
waterbodies may themselves represent a positive feedback by accelerating the thaw (Langer et al.
2016; Turetsky et al., 2020).

Since most of these potentially lost waterbodies are small and nameless ponds, it is hard

to point to specific cases, but the works cited above provide a number of telling examples. While
the focus in this context is negative water balance or loss of high-latitude waterbodies, this is
actually a widespread problem causing shrinking of many lakes. In Arctic areas, responses to
warming may differ substantially between perennial lakes and ephemeral wetlands, related to
ambient temperature and permafrost depth (Vulis et al. 2021). Although loss or gain of water
bodies does not classify as a tipping event in the very strict sense, i.e. there is not obvious strong,
self-reinforcing factors involved, it still is a climate driven change with potentially large,
widespread and irreversible consequences, and with repercussions on climate in terms of changes
GHG-emissions from vast areas.

*Feedbacks and tipping points*
Some essential feedbacks to climate change are involved in the change of Arctic waterbodies;
e.g. reduced ice and snow cover in the Arctic will promote further permafrost thaw. Certain
Arctic areas have experienced a major increase in breeding birds, notably gees, that promote
increased loadings of organic C and nutrients (Hessen et al. 2017). More organic carbon entering
water bodies from their terrestrial surroundings, combined with warming and eventually bird
induced eutrophication promotes GHG emissions (Wei et al. 2023).  It is important to make clear
that some of the impacts are contrasting, i.e. the loss of waterbodies may at first increase GHG
emissions (Keller et al. 2020; Paranaiba et al. 2021) but will eventually reduce GHG emissions.
Permafrost thaw and drainage of water-logged areas will increase $CO_2$-emissions but could
reduce $CH_4$-emissions. Sudden release of methane-hydrates upon permafrost thaw is a
possibility, yet hard to predict and quantify, and not specifically linked to aquatic habitats.

Few changes are as irreversible as complete habitat loss, and the climate-driven loss of

numerous water-bodies residing on permafrost over large geographical scales in Eurasia and
North America (due to permafrost thaw) with climate feedbacks in terms of changed GHG-
emissions is possible. In fact, as argued by Smol and Douglas (2007); "The final ecological
threshold for these aquatic ecosystems has now been crossed: complete desiccation". If strictly
adhering to the tipping point criteria as an event occurring when self-sustaining change in a
system is triggered beyond a forcing threshold, typically starting with positive feedback loops
and a runaway phase before finally the tipping-point brings the system into a different alternative
state, loss of waterbodies is not strictly a tipping point, but a binary shift. Abrupt permafrost
thaw, which can drive abrupt self-sustained formation or draining of thermokarst lakes, is
categorised as a "regional impact" climate tipping element by Armstrong McKay et al. (2022).
We extend this categorisation to include the lakes associated with these abrupt thaw processes,
seeing them as a coupled permafrost-lake systems with tipping dynamics involving both
components (Turetsky et al., 2020). Despite the scale considered here, the extent of open water
globally is relatively easy to quantify using remote sensing, and it is possible to make predictions
based on time-series and empirical relationships between temperature increase, permafrost thaw
and loss of water-bodies. Quantifying potential climate feedbacks related to processing of
organic C to $CO_2$ and $CH_4$ should be possible to predict within orders of magnitude, with initial
analysis suggesting abrupt thaw involving thermokarst lake formation and draining could double
the warming impact of gradual permafrost thaw (Turetsky et al., 2020).

**2.4. Switch from N to P-limitation**
Imbalance in biogeochemical cycles has become a major concern both on the local and global
scale. Anthropogenic emissions of $CO_2$ now appear as the major environmental challenge for
ecosystems and human well-being in the foreseeable future. In relative terms, however, the
anthropogenic effects on the global N-cycle are even more pronounced. Transformation of
atmospheric $N_2$ to more reactive reduced or oxidized forms of inorganic N by the fertilizer
industry and combustion processes has dramatically changed. Recent analyses of the global N-
cycle (Bodirsky et al. 2014; Zhang et al. 2020) suggest that various human activities currently
convert similar $N_2$ to total natural ecosystem fixation, and that both the use of N and P are far
beyond "safe boundaries" (Rockström et al. 2023).
Increased N-deposition may affect surface waters in fundamentally different ways. It will
increase the emissions of $N_2O$ (Yang et al. 2015), and increased deposition of inorganic N
promotes soil and water acidification through increased $NO_3$ in surface waters (Stoddard 1994).
It will, however, also affect elemental ratios in lakes and rivers (Hessen et al. 2009). The relative
proportions of these elements will determine the nature of elemental limitation for both
autotrophs and a range of heterotrophs, and could thus profoundly affect community composition
and ecosystem processes. One effect of such skewed inputs of N over P would be an intensified
P-limitation in surface waters or even large-scale shifts from N to P-limitation (Elser et al. 2009).
Rather than pinpointing specific lakes as examples, contrasting areas with high vs low N-
deposition Elser et al (2009) provide good regional examples, where Colorado offers examples
of regions with either high or low N-deposition, while southern regions of Norway and Sweden
experience up to ten-fold elevated levels of N-deposition compared with central regions.
Conversely, increased N-loss by denitrification, eventually associated with increased
internal P-loading may shift systems from P to N-limitation (Weyhenmeyer et al. 2007). Societal
implications include an increased prevalence of toxin producing cyanobacteria, purported to be
promoted in extent by warming (Paerl et al., 2008) and favouring non-N-fixing toxin producing
species where reduced-N concentrations are high relative to oxidized-N (Hoffman et al., 2022).
Additionally, a threshold on toxic effects on sensitive freshwater species has been proposed (i.e.
2 mg L$^{-1}$; Camargo et al., 2006; Moss et al., 2013), above which a marked decline in biodiversity
is expected.

*Feedbacks and tipping points:*
Changes in N- versus P-limitation of NPP are associated with changes in community structure,
both for the phytoplankton and macrophyte communities. While the shift from one limiting
nutrient to another represents a binary shift and abrupt transition, it is not driven by self-
propelling events or positive or negative feedbacks, since a shift from N- to P-limitation typically
is caused by N-deposition or agricultural use of fertilizers. While increased N-loading per se
could promote climate feedbacks in terms of N$_2$O, the switch from N to P-limitation or vice
versa is neither driven by climate nor does it exhibit strong feedbacks on climate. There is also
no inherent hysteresis, and when drivers change the system may immediately return to the other
limiting nutrient. For these reasons we do not classify this category as a tipping point according
to the definition above.

## 2.5. Salinization

Salinization is a prevalent threat to freshwater rivers, lakes and wetlands world-wide, particularly in arid and semi-arid regions and coastal areas. It is caused by a range of anthropogenic actions including water extraction, pollution and climate change (Herbert et al. 2015). The causes of salinization have historically been classified as being primary or secondary. Primary salinization refers to natural causes including wet and dry deposition of marine salts, weathering of rocks and surface or groundwater flows transporting salts from geological salt deposits. Secondary salinization refers to salinization caused by human activities such as irrigation with water rich in salts, rising of brackish and saline groundwater due to increased ground water extraction and increased seawater intrusion as a result of sea level rise. The distinction between natural and anthropogenic causes underlying salinization is becoming less clear cut due to climate change as anthropogenically caused changes in temperature, precipitation patterns and wind will affect the primary salinization processes (Oppenheimer et al. 2019). Salinization has severe consequences for aquatic communities (Jeppesen et al. 2015; Short et al. 2016; Cunillera-Montcusí et al. 2022). Salinization has a strong ecological impact often associated with osmotic stress and changes in biogeochemical cycles which often entails an increase in concentration of toxic sulfides (Herbert et al. 2015). In addition, studies focussing on the application of road salts indicate that salinization may disrupt lake water mixing and release of metals (Szklarek et al. 2022 and references therein). Negative effects of increased salinity have been described for trophic levels ranging from microorganisms to fish and birds (reviewed by Cunillera-Montcusí et al. 2022). In addition, salinization also has a high societal impact particularly related to domestic and agriculture water supply in arid and semi-arid regions (Williams et al. 1999).

A strong example on climate driven salinisation and its impact on biota is the long-term study (1938 – 2004) in two Canadian lakes (Sereda et al. 2011). Concomitant with periodic declines in precipitation, lake elevation declined and salinity increased in Jackfish and Murray lakes from 1938 to 2004. The increase in salinity caused an estimated 30% loss in diversity of macrobenthos. If salinity exceed thresholds where key species or functional groups are wiped out, it will no doubt represent an abrupt ecosystem transition, yet still not a tipping point in the sense that it is self-propelled.

*Feedbacks and tipping points*

Regime shift from clear to turbid may occur at 6-8 per mil salinity in systems with intermediate
to high nutrient loadings and have been associated with a change in zooplankton community
composition from cladocerans to more salinity tolerant cyclopoid copepods (Jeppesen et al
2007). Salinity induced regime shifts may also lead to dominance by microbial mats at the
expense of submerged macrophytes (Davis et al. 2003, Sim et al. 2006). While there are species-
specific tolerance thresholds to salinity, and these effects are expected to interact with other
stressors - including eutrophication (Jeppesen et al. 2007, Kaijser et al. 2019), color and turbidity
(Davis et al. 2003) - the process is not driven by feedbacks of increased salinization, but external
factors like warming, water (over)use and road salting. Hysteresis after refreshing of salinized
systems has been little studied but is likely strongly biogeochemical in nature as evidenced by
previously brackish waters that have been flushed with freshwater for over 90 years and still
contain high levels of chloride, sodium and sulfate (Van Dijk et al. 2019).

A weakened top-down control by zooplankton on phytoplankton occurring at moderate

high salinities would be an indirect consequence of salinization, leading to a worsening of
eutrophication symptoms (Gutierrez et al. 2018) and thus promoting indirect climate effects.
Salinization however tends to decrease $CH_4$ emissions (Herbert et al. 2015, Chamberlain et al.
2020, Gremmen et al. 2022). The decrease in $CH_4$ emission can be either caused by a decrease in
$CH_4$ production - e.g. because methanogens are outcompeted by sulfate reducers or are
negatively impacted by sulfide toxicity - or because an increase in methane oxidation (reviewed
by Herbert et al. 2015). The salinity induced decrease in aquatic $CH_4$ emissions may imply a
negative feedback with climate change, but only when this is not off-set by a decrease in carbon
burial. Insight in this balance is currently limited (Chamberlain et al. 2020), and while no doubt
salinization is widespread on regional scales and may reach threshold values for species and
processes, we do not categorize it is a tipping point under the cited criteria.

2.6. **Spread of invasive species**

Freshwaters are especially vulnerable to species loss and population declines as well as species
invasions due to their constrained spatial extent. Substantial ecosystem changes by reinforcing
interactions between invasive species and alternative states (i.e. macrophyte *versus*
phytoplankton dominance, as described above) may occur (Reynolds and Aldridge 2021). The
spread of several invasive species can change community composition and ecological functions
in dramatic ways, and can be regarded as sudden transitions with major site-specific or regional
impacts. Moreover, species invasions can be facilitated by climate change (Rahel and Olden,
2008), and notably flooding and other hydrological events can facilitate species invasion with
potentially far-reaching ecological consequences (Anufriieva and Shadrin 2018).

There are numerous examples of ecological consequences in lakes following species

invasions, and the major impacts of invasions by zebra mussel as well as the predatory
cladoceran *Bytotrephes* in the Great lakes, serve as striking examples of major impacts at the
regional scale even in very large lakes (Ricciardi and MacIsaac 2000). While species invasions
are of major ecological and societal concern, and can induce ecological tipping points in certain
lakes, they are generally not self-perpetuating involving internal feedbacks. No doubt it may be
appropriate to say that invaded system may cause irreversible changes or hysteresis in specific
lakes or lakes within regions.

*Feedbacks and tipping points*
Climate, both in the context of warming that open for latitudinal and altitudinal spread of species
(Hessen et. al. 2006) and hydrological events that likewise may promote invasions (Anufriieva
and Shadrin 2018) may pose drastic changes in community composition and ecosystem functions
to an extent that qualify as abrupt shifts. Species invasions may also interact with other drivers
lowering the potential thresholds (of nutrients, temperature, browning, etc.) for a shift to occur,
and vice versa, by impacting on previously occurring stabilizing mechanisms (Willcock et al.
2023). Likewise, species shifts may have repercussions on GHG-emissions. We do not pursue
the discussion feedbacks and potential tipping points further for this candidate category,
however, since we have constrained our definition of tipping points to situations with internal
feedback and regional occurrence. Given the widespread anthropogenic changes promoting
invasive species in aquatic communities worldwide, the often abrupt and unpredictable shifts that
may follow from this deserves further attention.
**3. Discussion**
Freshwaters are one of the most vulnerable ecosystems and resources globally and will
increasingly be so in warming world. They also link catchment properties and terrestrial changes
to marine systems, and notably lakes serve as good sentinels of global change (Adrian et al.
2009). Population declines and species loss in freshwaters are happening at an alarming pace
underpinning the urgency for evidence of ecological tipping points in response to environmental
change. Drinkable freshwater is a scarce resource both in terms of water quality and availability
(Yao et al. 2023). Predicting (and preventing) sudden shifts in water quality and quantity is
therefore a high priority also from an anthropocentric perspective, and insights into feedbacks,
thresholds and tipping points are highly relevant to lakes. Lakes are also major players in the
global climate, and besides being highly vulnerable to climate change, they can provide strong
feedback to the climate by ventilating a substantial share of terrestrially fixed C back to the
atmosphere as $CO_2$ and $CH_4$ (Cole et al. 2007; Tranvik et al. 2009; Raymond et al. 2013). Lakes
are also subject to changes, sometimes sudden, due to climate change and other natural or
anthropogenic drivers. In fact, some of the first and most striking examples on tipping points and
regime shift come from lake studies (Scheffer et al. 1993; Jeppesen et al. 1998).

We argue that there are two key drivers that may shift lakes towards major ecological

changes, as well as increased climate feedback by GHG emissions, namely eutrophication and
browning. Both these drivers are promoted by warming, which may be seen as a separate driver.
Both processes are also characterised to some degree by self-sustaining feedback loops, feedback
to climate in terms of GHG-emissions, and are also strongly integrated with land surface impacts
in the catchment (Fig. 3). Warming, browning, and eutrophication lead to increases in
stratification, heterotrophy, and phytoplankton or macrophyte mass, which collectively drive
benthic oxygen depletion and in turn increased GHG emissions (helping to drive further
warming and DOM loading from land) and internal P loading (driving further eutrophication)
(Meerhoff et al. 2022). Several of these processes can feature tipping points (eutrophication and
potentially DOM loading), which warming will likely make easier to reach.

Few processes have been more thoroughly described in terms of drivers, impacts and

remedies than freshwater eutrophication. The drivers are well known (nutrient loadings from
agricultural activities and locally also from sewage discharges), despite long-term controversies
regarding the relative importance of N or P in promoting eutrophication (e.g., Smith & Schindler
2009, Paerl et al. 2016). There are also long traditions for predictive hydraulic models that link
the load of P to algal blooms and benthic $O_2$-depletion (e.g., Vollenweider type models, Imboden
1974). Moreover, given the scarcity, increasing demands and increasing prices of P as a
commodity worldwide, there are indeed strong arguments to close the loop for P and reduce
losses to the environment (Brownlie et al. 2022). Due to the strong impact of $O_2$-depletion on
sediment release of P and thus internal fertilization (Soendergaard et al. 2002), that will play in
concert with food-web driven feedbacks (cf. Fig. 3), tipping points in this context can be
identified, while the climate component is difficult to separate.

Browning shares many of these attributes in terms of increased net heterotrophy. The

shift from net autotrophy with a net uptake of $CO_2$ to net heterotrophy with a net release of $CO_2$
(plus $CH_4$) also represents a binary situation. However, since most boreal lakes are already net
heterotrophic owing to microbial conversion of organic C (Hessen et al. 1990; Cole et al. 1994;
Larsen et al. 2011), most boreal lakes simply become more heterotrophic, hence there is no
tipping point in this context. However, an increased degree of heterotrophy combined with
increased thermal stability, will promote deep-water anoxia, thereby internal P-cycling and
GHG-release. Since the key driver here is the external load of terrestrial DOM, the feedback
component driving P-release is weaker than in the case of eutrophication. Nevertheless, an
increased release of GHGs no doubt poses a feedback to the climate and hence the terrestrial
systems that may promote further browning. These processes are amplified by climate change,
and have global consequences in terms of GHG emissions. Given the high confidence in this
case we recommend it as apriority for parametrization of models to underpin future predictions
of the impacts of global tipping points in lakes on GHG emissions.


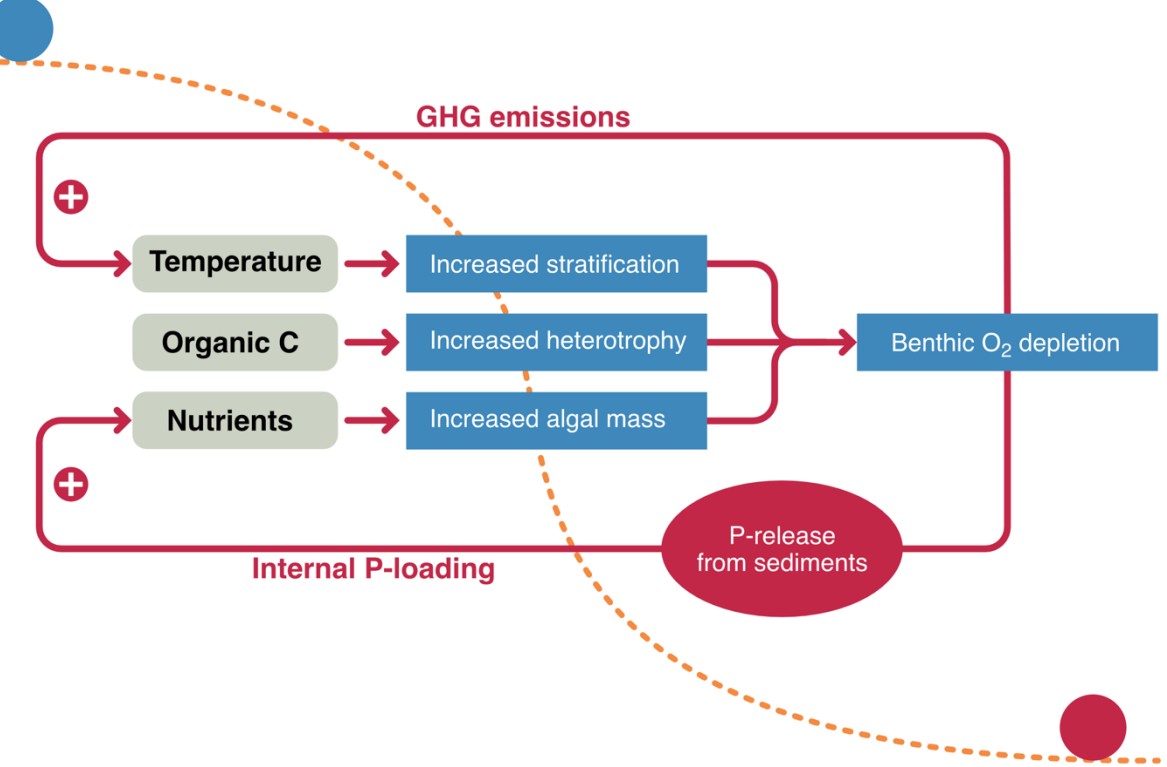

Fig. 3. The interactive role of eutrophication, DOM-export (browning) and warming on lakes. Separately or combined they promote benthic $O_2$-depletions which cause an internal feedback by P-loading from sediments and a climate feedback via release of greenhouse gases. The potential shift between states (blue to red circle) is indicated.

As a separate type of binary tipping point which is likely to be widespread and related to GHG-release, we propose the loss of water bodies, notably Arctic ponds. This is driven by permafrost thaw in the case of thermokarst-linked lake formation or disappearance (categorised as a regional tipping element in previous assessments (Armstrong McKay et al. 2022). Together, the coupled permafrost-lake system can act as a localised tipping system with the lake providing key feedbacks to help drive self-sustaining thaw. This makes the tipping points easy to monitor (by remote sensing), and predictable in the sense that it will be closely linked to permafrost thaw. There are, however, feedbacks to the climate, with potentially high emissions during the drying process (Marcé et al. 2019; Turetsky et al. 2020). A different situation would be the less widespread case of new waterbodies formed by collapsing palsas, in cases of retreating glaciers. Given the potential scale of occurrence, the net effect of permafrost thaw and increased release

of $CO_2$ balanced with the effect of disappearing waterbodies and potential changes in net GHG
emissions requires further attention as a matter of high priority.

3.1.Gradients or tipping points – does it matter?

One could argue that what matters is whether a change or process is linear (and thus more
predictable) or non-linear (and less predictable), and that the rest is semantics. This is truly not
the case, since there are substantial differences in what is considered as a 'tipping point', not the
least in terms of whether impacts are easily reversible or are effectively "locked in" (e.g.
hysteresis). Still, from an ecosystem perspective, abrupt shifts, even if they do not qualify as
tipping points, may have devastating effects that should urge us to invest more in preventing
deterioration as we do not know where/if a sudden shift may occur. As argued by Moss et al.
(2008): "the sort of precision demanded by legislators and lobbies will never be attainable and
this has been a major weapon used to delay regulation of agricultural activities."

Shifts between ecological states do not necessarily involve alternative stable states with

hysteresis. In fact, both the concepts of abruptness and irreversibility depends on one's
perspective of time. Over a lakes life-time, shifts back and forth between states occurring over
years or even decades are "sudden" relative to the human lifespan. For example, Rühland et al.
(2008) report apparent coherence in diatom community shifts post 1850 on hemispheric scales
over 100 years or so. Similarly, a coherent, global increase in hypoxia in lakes has been reported
over a 100 year period (from about 1850) by Jenny et al. (2016). Likewise regional patterns of
species turnover (β-diversity) over 200 years demonstrated regional differences in species
turnover, but also recent changes attributed to warming (Kahlert et al. 2020). As warming
progresses, such studies forms good baselines for future changes. If the observational time step is
increased to centuries, then it is likely that more large-scale examples will come through in
paleo-studies. In fact, there are several examples on coherence in lake responses to climate
variability or climate change, some of which take place over short time spans (Stone et al. 2016;
Isles et al. 2023). Finally, multiple drivers may jointly push lakes towards shifts or tipping
points, as shown in Huang et al. (2022) and Willcock et al. (2023).

Taken together, there are at least two major reasons why an improved understanding of

sudden changes in lake ecosystems are imperative; they are highly vulnerable to climate change
and other anthropogenic stressors globally, and they serve as major feedbacks to the climate
system through GHG emissions. Being well-mixed and semi-closed entities that reflect changes
in catchment properties, they also serve as sentinels of global change (Adrian et al. 2019). For
fresh waters in general, lakes are crucial in the hydrological cycling, and link the terrestrial and
marine ecosystems. The major tipping point dynamics converge in oxygen depletion, primarily
in deeper strata and the sediment surface, which promotes feedbacks and hysteresis in terms of
internal P release as well as increased GHG-emissions. High nutrient load, increased inputs of
dissolved organic C and warming all drive oxygen depletion, and while many problems related
to global warming boil down to the obvious recommendation of reduced use of fossil fuel and
other GHG-emitting activities, reducing nutrient use and losses to within the carrying capacity of
the system (i.e. from ecosystem to global scales) is comparatively simpler both for N and P when
compared to C (Rockström et al. 2009; 2023). The incentives should be even greater for closing
the P-loop, given the potential for scarcity of this non-substitutable element and its role in lake
eutrophication (Brownlie et al. 2022).
Regime shifts and tipping points are concepts closely linked to resilience (Andersen et al.
2008; Spears et al. 2017). Lakes represent excellent model case studies in this respect and have
been used widely to demonstrate theories of ecological stability and resilience that are needed to
underpin preventative management approaches and to guide science-based environmental policy.
The full importance of the vulnerability of lakes to climate and other anthropogenic impacts, as
well as their feedback to climate is not yet fully acknowledged, so there is a need both for
science and communication in this regard. However, we argue that the search for empirical
evidence to underpin theory should not prevent societies and managers taking more action to
protect fresh waters in the meantime.

**4. Conclusions**
Anthropogenic forcing may induce non-linear, abrupt changes in freshwater ecosystems, and
awareness of such potential threshold effects are important. Here we focus on lake and pond
systems that may be subject to tipping points, where self-reinforcing feedback and some type of
post-change hysteresis can be identified. To qualify in this context these changes must also be
relevant or larger scales (i.e. regions or biomes), or a large number of systems. Two types of
potential tipping points were identified based on these criteria; eutrophication and browning. The
first is an example of a widespread phenomenon, the second occurring in lakes in the boreal
biome. In both cases, climate in involved as a driver, and the changes in terms of deep-water
anoxia and internal P-cycling may boost the emission of GHG-gases from such systems. While
not tipping points according to the criteria applied here, two types of binary shifts is also
discussed; loss of water-bodies, notably in the Arctic areas, caused by permafrost thaw or
negative water balance as well as shifts between N and P-limitation caused by N-deposition.
Notably the first is driven by climate change and will also have repercussions on climate by
changes in GHG-emissions. Finally, salinization and species invasions are also changes that may
occur over large scales with potentially abrupt ecosystem changes, and where changes in
temperature and precipitation as important drivers.

*Acknowledgement*s: This work has benefitted from discussions withing the group behind the
Tipping Point report presented at COP 28 meeting in Dubai, and the ms has benefitted
substantially from inputs and suggestions by John Smol and an anonymous reviewer on the first
draft.

*Author contribution*: DH conceived the idea and wrote the first draft, all authors have contributed
to the writing and approved the final manuscript.
*Code and data availability*: Not relevant
*Competing interests*: None.
*Special issue statement*: Part of the ESD tipping points special issue

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
