# Peer review of "Lake ecosystem tipping points and climate feedbacks"

_Earth System Dynamics, 2023_

## Referee Comment (RC2)

**Review – Lake ecosystem tipping points**

Dag Hessen et al. Earth System Dynamics     Jan 5, 2024

General comments

I realize this review is over 5 single-spaced pages, but the vast majority of the comments are minor.

This paper is certainly of broad interest and is well-written but please see the detailed comments below. Some sections could be improved.  For example, there are many extremely long, complex sentences that may be difficult to follow. I made some suggestions for many (but not all) of these.

After examining six potential drivers of lake tipping points, the authors found that only two potential drivers of tipping points (browning and eutrophication) fit their criteria and that these two are related or exacerbated by climate change. My main comment here is that this does not really fit the title of the paper "Lake ecosystem tipping points and climate feedbacks". Although they do mention that they focus on tipping points that are not necessarily driven by climate change per se (line 124) and acknowledge that climate can also be seen as separate driver (line 502), I wonder why the authors have not included situations where lakes that can cross tipping points that are mainly by triggered by accelerated warming... without needing additional stressors of eutrophication or browning (e.g. many High Arctic and very remote water bodies on granitic bedrock etc.). I certainly don't disagree that both browning and eutrophication are major drivers of regime shift changes in lakes and that climate change can help further push waterbodies towards a tipping point. However, and my main point here, waterbodies can also reach an ecological tipping point with accelerated warming and in the absence of these two drivers. The biggest take-away from this paper may appear to be that tipping points can only be crossed if lakes are affected by these two primary drivers (browning and eutrophication). Perhaps this needs to be better clarified and elaborated upon at the beginning of the paper (introduction) and again in discussion. I appreciate that Table 1 provides the "events" that trigger tipping points and the authors have examined all of these and that climate change is not considered an "event" here. Of course, recent accelerated warming is mainly anthropogenic and these feedbacks are important --  but I guess I am having some difficulty in seeing why climate is not considered as a main driver- particularly for lakes in remote regions (high latitude, parks etc.) where many studies have shown regime shifts in response to warming and in the absence of these two drivers. Maybe some further explanation is in order?

The section on disappearance of waterbodies (starting on line 324) seems to be solely focused on permafrost thaw as the component. As I detail below, when they talk about Smol and Douglas 2007 (lines 359-369-ish and again on lines 544-554-ish) and disappearing lakes– it seems they are incorrectly interpreting how these High Arctic waterbodies on granitic bedrock are disappearing. They intimate (unless I am misreading this section) that these small ponds

have disappeared because of *permafrost thaw and draining* (i.e. thermokarst lake). Contrary to what the authors indicate in this section, Smol and Douglas (2007) show that water levels in ponds such as those on Cape Herschel are not similarly influenced by permafrost drainage (as some subarctic waterbodies) and represent a more direct link to temperature, precipitation etc. They are excavated in granite and their disappearance was shown to be due to higher evaporation:precipitation. I provide some details below.

The authors argue that the disappearance of these waterbodies should not be considered "tipping points" per se but a binary shift (line 367). Could the authors elaborate on this point here?

The section on gradients or tipping points (line 559) is an important addition as it helps clarify the importance of abrupt changes, regardless of whether a set of criteria deems the change to qualify as a "tipping point" or not. The time perspective is important.

In the sections on TOC and browning etc – Well, clearly this is a paleolimnologist who is reviewing this paper(!), but the authors might consider the concept of "re-browning", as we have argued, given the perspective of several centuries of paleo inferences of DOC data. Perhaps lakes are returning to their natural "browner" conditions (i.e. re-browning).

If interested, some of these papers are:

Meyer-Jacob, C. et al. 2020. Re-browning of Sudbury (Ontario, Canada) lakes now approaches pre-acidification lake-water dissolved organic carbon levels. Science of the Total Environment 725: 138347.

Meyer-Jacob, C. et al. 2019. The browning and re-browning of lakes: Divergent lake-water organic carbon trends linked to acid deposition and climate change. Scientific Reports 9: 16676

The section on Salinization – I think is fine, but the authors could mention the threat of road salt seepage – which we know is changing lakes.

In the context of this paper, the section on the spread of invasive species seems out of place – why include this if the authors deemed this not to be a candidate for tipping points? It is short and, in my opinion, not very informative.

Detailed Comments

Perhaps of relevance to this paper is a J. Paleolimnology  review paper on regime shifts in lakes in response to climate change and anthropogenic activities (Randsalu-Wendrup et al. 2016). https://link.springer.com/article/10.1007/s10933-016-9884-4

**Line 32:** as noted later, maybe add  "or higher precipitation to evaporation ratios" after "shifts"

**Line 34:** remove  "on"

**Line 37** – for parallel structure, should be "increase" nt "increased"

**Line 39:** Several of these processes can feature potential tipping point thresholds, which further warming will likely make easier to  surpass.

**Line 56:** is "populous" the correct word here?

**Line 57** and elsewhere – usually convention is now sub-Arctic – with Arctic always uppercase

**Line 58-59**: oddly worded. Perhaps change to

Widespread loss of waterbodies, from Arctic or sub-arctic ponds to wetlands or bogs might qualify as one type of tipping point, but are not self-propelled by internal feedbacks *per se,* but rather  by permafrost thaw (Smol and Douglas 2007).

Also, in this instance our 2007 PNAS paper is not appropriate for permafrost thaw, as these Cape Herschel ponds were excavated in granite and the loss of water was higher evap:precip ratios.  Although we published papers and reviews discussing loss of ecosystems with permafrost thaw, the Smol and Douglas 2007 PNAS paper is on ponds excavated in granite bedrock (so they are like bathtubs in the bedrock – not permafrost) – chosen as not directly influenced by permafrost thaw but increased evaporation. We had pondwater conductivity measures going back to 1983 – so we could show it was evaporation.  So, these ecosystems are disappearing due to evaporation, not permafrost thaw.

So it would be correct to say   by higher evaporation to precipitation ratios (Smol and Doulgas 2007) as well as permafrost thaw (many papers can be cited here…..  Chapter 7 of my new book Smol, J.P. 2023. *Lakes in the Anthropocene: Reflections on tracking ecosystem change in the Arctic.*  Excellence in Ecology Book Series, International Ecology Institute (ECI), Oldendorf/Luhe, Germany. 13 chapters. 438 pp. – has many examples.  One common one is:  Smith, L. C., Sheng, Y., MacDonald, G. M., and Hinzman, L.D. 2005. Disappearing Arctic lakes. Science 308: 1429,

**Line 134**: add "a" before "tipping point"
**Line 142**: hysteresis should be plural "hystereses"
**Line 148**: remove "if not"; change "impacts" to "stressors"; remove "s" from "waters"

**Line 153**: change "among" to "as well as"

**Line 155**: change "pointed" to "identified"

**Line 160**: figure caption within the brackets: I found this hard to follow. Should "drive" here not be "driven"? "(...a tipping point-driven self-sustaining change)"?

Lines 165-169: this is an overly complex sentence that is hard to follow. Could you simplify/clarify?

Hysteresis can be strengthened by eutrophication-driven biological changes  in fish composition and size structure  that have cascading effects on zooplankton and phytoplankton as well as strong impacts  on fish-mediated nutrient cycling (Brabrand et al. 1990). This in turn,  will maintain a system with deepwater anoxia and high nutrient load, supporting the release of GHGs (Fig. 2).

**Line 175:** Replace "speed" with "an acceleration"

**Line 176:** increases thermal stability and the duration and strength of stratification

**Line 177:** Minor, but there is a newer Woolway et al review that might be appropriate here: Woolway et al. 2022. Lakes in hot water: the impacts of a changing climate on aquatic ecosystems. BioScience 72: 1050-1061

**Line 187**: add a hyphen to "eutrophication-induced"

**Lines 186-190**: This is a very long and complex sentence – perhaps split in two.

**Line 201** (and elsewhere): Yang et al. (2015) is missing from the reference list.

**Lines 210-212**: Why only shallow lakes?

**Line 213:** what is meant by "coherent tipping"? Do you mean coherent "threshold exceedance"?

**Lines 215-218**: This sentence was long and complicated and I found it difficult to follow. How about:

However, given the dearth of studies that generate bi-directional carbon flux data to assess the balance between emission and burial in lakes, it remains unknown whether the effect of  climate feedback  can be buffered by the projected eutrophication-driven increases in lake carbon burial (Anderson et al.2020).

**Line 221:** Consider starting a new sentence after (Grasset et al. 2020).

**Line 236**: There are several other studies relevant to treeline shifts and DOC or TOC including:

Pienitz, R. et al. 1999. Paleolimnological reconstruction of Holocene climatic trends from two boreal treeline lakes, Northwest Territories, Canada. *Arct, Antarct., and Alpine Res.* 31:82-93. https://doi.org/10.1080/15230430.1999.12003283

Rühland, K.M et al. Limnological characteristics of 56 lakes in the Central Canadian Arctic Treeline Region. J. Limnol. 62:9-27.

**Lines 247-250**: can you provide a reference for this. Also, the closed bracket is missing at end of sentence.

**Line 309**: delete "as of"

**Lines 310-312:** This sentence was difficult to understand as written. How about changing to:

Given that high concentrations of DOM and deep-water anoxia are common, most  boreal lakes are net heterotrophic and thus conduits of CO2, and often  CH4

**Lines 313-315**: This long sentence would be clearer if split into two.

If it eventually leads to oxygen depletion and cascading feedbacks then it would qualify as a tipping point.  However, there would be a time delay between the two events,  with the latter  being the critical tipping event.

**Line 327**: should this not be "drought" rather than "draught"?

**Line 329**: usually High Arctic is capitalized (but perhaps depends on the journal).

**Line 330**: delete "and" before "onset" and replace with "further promoting the onset of permafrost thaw..."

**Line 331**: delete "both" – the sentence starts with "Both"

**Line 339**: change "share" to "sheer"

**Line 344**: as noted above the Smol & Douglas paper is on evaporation – so I would change the sentence to:  While the main problem is loss of water bodies affected by warming-induced increased evaporation rates (Smol and Douglas, 2007) and permafrost thaw (maybe cite Smith et al., 2005 here) ….

**Line 345-347**: In addition to "collapsing palsas and thermokarst areas" another climate-mediated phenomenon related to permafrost thaw is retrogressive thaw slumps that increase inorganic sediments to freshwater systems and affect biodiversity (Thienpont et al. 2013. Freshwater Biology; Heino et al. 2020).

**Lines 351-353**: The sudden introduction of "bird induced eutrophication" is a bit odd. Can you introduce/connect this better to the rest of this section – it seems to come out of nowhere. Also bird-induced here should be hyphenated.

**Line 361**: North America should not be hyphenated.

**Line 372**: delete extra period after reference.

**Line 461**: change "is" to "as"

**Line 497**: add an "s" to "regime shifts"

**Line 522 on**:  You might consider discussing "re-browning" here as well.

**Near lines 572 to 575**:  There is also the Smol et al 2005 PNAS compilation across the circum-polar Arctic that perhaps you meant to cite here?  It is in the reference list, but not actually cited in text as far as I can see… Also, the newer Kahlert et al. (2022) paper would be appropriate here: Kahlert, M. et al.  2022. Biodiversity patterns of Arctic diatom assemblages in lakes and streams: Current reference conditions and historical context for biomonitoring. Freshwater Biology 67: 116-140.

Interesting paper.

---

## Author Comment (AC1)

**Response to Reviewers for Hessen et al., Lake ecosystem tipping points and climate feedbacks**

*We thank the reviewers for their positive judgment on our ms, and the very constructive comments. We outline below revisions in response to the general and specific points raised by each reviewer. We thank the reviewers for the suggestion to broaden the scope of the paper. We acknowledge that the original submission followed a definition of 'tipping point' applied in the report, 'Global Tipping Points' (Lenton et al., 2023) presented at the recent COP28-meeting (which is now cited); namely that there should be self-reinforcing feedbacks and evidence of hysteresis (as explained also in the ms). We have now explained this rationale in the manuscript along with considerations of broader contexts as recommended by the reviewers. A number of new references have been added to support our conclusions, and also to provide examples as suggested by RC1. The revisions are more than those pasted in the responses below, but the major revisions directly addressing suggestions or requests are refereed to with line number and revised text below.*

**Reviewer 2 (RC2)**:

**Rev 2 General comments**

I realize this review is over 5 single-spaced pages, but the vast majority of the comments are minor.

This paper is certainly of broad interest and is well-written but please see the detailed comments below. Some sections could be improved. For example, there are many extremely long, complex sentences that may be difficult to follow. I made some suggestions for many (but not all) of these.

After examining six potential drivers of lake tipping points, the authors found that only two potential drivers of tipping points (browning and eutrophication) fit their criteria and that these two are related or exacerbated by climate change. My main comment here is that this does not really fit the title of the paper "Lake ecosystem tipping points and climate feedbacks". Although they do mention that they focus on tipping points that are not necessarily driven by climate change per se (line 124) and acknowledge that climate can also be seen as separate driver (line 502), I wonder why the authors have not included situations where lakes that can cross tipping points that are mainly by triggered by accelerated warming… without needing additional stressors of eutrophication or browning (e.g. many High Arctic and very remote water bodies on granitic bedrock etc.). I certainly don't disagree that both browning and eutrophication are major drivers of regime shift changes in lakes and that climate change can help further push waterbodies towards a tipping point. However, and my main point here, waterbodies can also reach an ecological tipping point with accelerated warming and in the absence of these two drivers. The biggest take-away from this paper may appear to be that tipping points can only be crossed if lakes are affected by these two primary drivers (browning and eutrophication). Perhaps this needs to be better clarified and elaborated upon at the beginning of the paper (introduction) and again in discussion. I appreciate that Table 1 provides the "events" that trigger tipping points and the authors have examined all of these and that climate change is not considered an "event" here. Of course, recent accelerated warming is mainly anthropogenic and these feedbacks are important -- but I guess I am having some difficulty in seeing why climate is not considered as a main driver- particularly for lakes in remote regions (high latitude, parks etc.) where many studies have shown regime

shifts in response to warming and in the absence of these two drivers. Maybe some further explanation is in order?

*Response: We thank the reviewer for the insightful comments. We definitely agree that climate change per se in many cases is the key driver towards potential tipping points, not only in concert with browning and eutrophication. Please note comment at outset of this responses document on the definition of 'tipping points' that informed our analysis. Since we have broadened the scope to include discussion of other type of abrupt shifts of non-linear changes, related to climate change, we agree that it is logical to place more emphasis on warming as a specific driver. This holds especially for the binary shift from presence/absence of waterbodies. In this context we agree that negative water balance per se, irrespective of permafrost thaw, can occur both at high latitudes and elsewhere (se response below).*

*We pondered quite a bit on the title, bur arrived at this which we still believe covers the topic quite well whilst being accessible to a wide audience. The title explicitly refers to feedbacks to climate through GHG-emissions, irrespective of the driver and whether or not the changes involve self-reinforcing feedbacks.*

*The point of warming as a driver is addressed already in the revised Abstract: We identify systems and drivers that could lead to self-sustaining feedbacks, abrupt changes and some degree of resilience, as opposed to binary states not subject to self-propelling changes or resilience. Changes driven by warming, browning, and eutrophication can cause increased lake stratification, heterotrophy (browning), and algal mass (eutrophication), which separately or collectively drive benthic oxygen depletion, internal phosphorus-loading and in turn increase greenhouse gas (GHG) emissions. Several of these processes can feature potential tipping point-thresholds, which further warming will likely make easier to surpass.*

The section on disappearance of waterbodies (starting on line 324) seems to be solely focused on permafrost thaw as the component. As I detail below, when they talk about Smol and Douglas 2007 (lines 359-369-ish and again on lines 544-554-ish) and disappearing lakes– it seems they are incorrectly interpreting how these High Arctic waterbodies on granitic bedrock are disappearing. They intimate (unless I am misreading this section) that these small ponds have disappeared because of *permafrost thaw and draining* (i.e. thermokarst lake). Contrary to what the authors indicate in this section, Smol and Douglas (2007) show that water levels in ponds such as those on Cape Herschel are not similarly influenced by permafrost drainage (as some subarctic waterbodies) and represent a more direct link to temperature, precipitation etc. They are excavated in granite and their disappearance was shown to be due to higher evaporation:precipitation. I provide some examples below, but we have revised the text in accordance with these comments throughout.

*Response: This is indeed a good point. We completely agree, and have revised the ms accordingly throughout, e.g. L 59-66: Some types of changes can be classified as binary, i.e. either-or situations at the system level. Increased temperature and/or reduced precipitation may induce negative water balance and shrinking of water volumes to the level where lakes or ponds simply disappear. Many lakes worldwide are facing reduced water volumes, but perhaps most striking is the widespread loss of high-latitude waterbodies, from Arctic or sub-Arctic ponds to wetlands or bogs. Such phenomena may qualify as one type of tipping point, but are not self-propelled by internal feedbacks per se, but rather by higher evaporation to precipitation ratios (Smol and Douglas 2007) or permafrost thaw (Smith et al. 2005; Webb et*

*al. 2022; Smol 2023).  L 376-390: While the main problem is loss of water bodies affected by warming-induced increased evapotranspiration rates (Smol and Douglas 2007) and permafrost thaw (Smith et al. 2005), there are also cases where collapsing palsas and thermokarst areas create new waterbodies, and these waterbodies may themselves represent a positive feedback by accelerating the thaw (Langer et al. 2016; Turetsky et al., 2020).*

*Since most of these potentially lost waterbodies are small and nameless ponds, it is hard to point to specific cases, but the works cited above provide a number of telling examples. While the focus in this context is negative water balance or loss of high-latitude waterbodies, this is actually a widespread problem causing shrinking of many lakes. In Arctic areas, responses to warming may differ substantially between perennial lakes and ephemeral wetlands, related to ambient temperature and permafrost depth (Vulis et al. 2021). Although shrinking, appearance or loss of water bodies does not classify as a tipping event in the very strict sense, i.e. there is not obvious strong, self-reinforcing factors involved, it still is a climate driven result of climate change with potentially large, widespread and irreversible consequences.*

The authors argue that the disappearance of these waterbodies should not be considered  "tipping points" per se but a binary shift (line 367). Could the authors elaborate on this point here?

*Response: We have discussed this distinction later in the ms, but agree that this is a critical distinction that should be presented early on. Also, we now elaborate more on abrupt shifts (like loss of water bodies), and thus have elaborated both this and the distinction between binary shifts and tipping points.*

The section on gradients or tipping points (line 559) is an important addition as it helps clarify the importance of abrupt changes, regardless of whether a set of criteria deems the change to qualify as a "tipping point" or not. The time perspective is important.

*Response: Yes, and this is related to the point above. We have also elaborated on the time perspective.*

In the sections on TOC and browning etc – Well, clearly this is a paleolimnologist who is reviewing this paper(!), but the authors might consider the concept of "re-browning", as we have argued, given the perspective of several centuries of paleo inferences of DOC data.  Perhaps lakes are returning to their natural "browner" conditions (i.e. re-browning).

If interested, some of these papers are:

Meyer-Jacob, C. et al. 2020. Re-browning of Sudbury (Ontario, Canada) lakes now approaches pre-acidification lake-water dissolved organic carbon levels. Science of the Total Environment 725: 138347.

Meyer-Jacob, C. et al. 2019. The browning and re-browning of lakes: Divergent lake-water organic carbon trends linked to acid deposition and climate change. Scientific Reports 9: 16676

*Response: Indeed, this is an important and ongoing debate and we thank the reviewer for highlighting these papers. With insufficient time series of water quality, sediment analysis no doubt gives key insights. This is also dependent on the time perspective and the causes of browning. Nevertheless, we agree that "re-browning" definitely should be considered and covered – and is now included in the revised ms. E.g. L331-334: The temporal aspect also deserves further attention. If the main source of browning is afforestation, responses will proceed slowly compared with cases where reduced acid deposition is the main driver, yet both drivers operate on decadal timescales. In the latter case, the browning could represent a re-browning (Meyer-Jacob et al. 2020).*

The section on Salinization – I think is fine, but the authors could mention the threat of road salt seepage – which we know is changing lakes.

*Response: While not directly climate driven, road salting is no doubt a relevant contributor to salinization in many areas (some of us have also worked on this topic), and this aspect is now covered. Actually road-salting was originally included in an earlier version, and is now reintroduced: 484-487: In addition, studies focussing on the application of road salts indicate that salinization may disrupt lake water mixing and release of metals (Szklarek et al. 2022 and references therein). Negative effects of increased salinity have been described for trophic levels ranging from microorganisms to fish and birds (reviewed by Cunillera-Montcusí et al. 2022).*

In the context of this paper, the section on the spread of invasive species seems out of place – why include this if the authors deemed this not to be a candidate for tipping points? It is short and, in my opinion, not very informative.

**Response: We had some discussions on whether or not to include this point when developing the original analysis. However, since climate change can be a major cause of species spread and invasions we felt it should be included. In some cases the arrival of an invasive species dramatically changes the system to the extent that there is a "before-and-after" situation. Hence, since our intention was to discuss a range of abrupt changes before focusing on tipping point cases according to the criteria set in the overarching definition, we decided to include invasions, but have extended the text on this category substantially for context, L521-552: Spread of invasive species**
*Freshwaters are especially vulnerable to species loss and population declines as well as species invasions due to their constrained spatial extent. Substantial ecosystem changes by reinforcing interactions between invasive species and alternative states (i.e. macrophyte versus phytoplankton dominance, as described above) may occur (Reynolds and Aldridge 2021). The spread of several invasive species can change community composition and ecological functions in dramatic ways, and can be regarded as sudden transitions with major site-specific or regional impacts. Moreover, species invasions can be facilitated by climate change (Rahel and Olden, 2008), and notably flooding and other hydrological events can facilitate species invasion with potentially far-reaching ecological consequences (Anufriieva and Shadrin 2018).*
*        There are numerous examples of ecological consequences in lakes following species invasions, and the major impacts of invasions by zebra mussel as well as the predatory*

*cladoceran Bytotrephes in the Great lakes, serve as striking examples of major impacts at the regional scale even in very large lakes (Ricciardi and MacIsaac 2000). While species invasions are of major ecological and societal concern, and can induce ecological tipping points in certain lakes, they are generally not self-perpetuating involving internal feedbacks. No doubt it may be appropriate to say that invaded system may cause irreversible changes or hysteresis in specific lakes or lakes within regions.*

*Feedbacks and tipping points*
*Climate, both in the context of warming that open for latitudinal and altitudinal spread of species (Hessen et. al. 2006) and hydrological events that likewise may promote invasions (Anufriieva and Shadrin 2018) may pose drastic changes in community composition and ecosystem functions to an extent that qualify as abrupt shifts. Species invasions may also interact with other drivers lowering the potential thresholds (of nutrients, temperature, browning, etc.) for a shift to occur, and vice versa, by impacting on previously occurring stabilizing mechanisms (Willcock et al. 2023). Likewise, species shifts may have repercussions on GHG-emissions. We do not pursue the discussion feedbacks and potential tipping points further for this candidate category, however, since we have constrained our definition of tipping points to situations with internal feedback and regional occurrence. Given the widespread anthropogenic changes in aquatic communities worldwide, the often abrupt and unpredictable shifts that may follow from this deserves further attention.*

**Rev 2 Detailed Comments**

*Response: We thank the reviewer for taking the time to provide such a comprehensive assessment. We have addressed all detailed comments listed below by implementing suggested changes in the revised ms.*

Perhaps of relevance to this paper is a J. Paleolimnology review paper on regime shifts in lakes in response to climate change and anthropogenic activities (Randsalu-Wendrup et al. 2016). https://link.springer.com/article/10.1007/s10933-016-9884-4

**Line 32:** as noted later, maybe add "or higher precipitation to evaporation ratios" after "shifts"

**Line 34:** remove "on"

**Line 37** – for parallel structure, should be "increase" nt "increased"

**Line 39:** (SEE ATTACHED PDF --- MY CROSS-OUTS AND EDITS ARE NOT SHOWING IN PASTED WORD DOC) Several of these processes can feature potential tipping points thresholds, which

further warming will likely make easier to reach surpass.

**Line 56:** is "populous" the correct word here?

**Line 57** and elsewhere – usually convention is now sub-Arctic – with Arctic always uppercase

**Line 58-59**: oddly worded. Perhaps change to    (SEE ATTACHED PDF ---  MY CROSS-OUTS AND EDITS ARE NOT SHOWING IN PASTED WORD DOC)

Widespread loss of waterbodies, from Arctic or sub-arctic ponds to wetlands or bogs might

qualify as one type of tipping point, but are not self-propelled by internal feedbacks *per* se,themselves but rather than by permafrost thaw (Smol and Douglas 2007).

Also, in this instance our 2007 PNAS paper is not appropriate for permafrost thaw, as these Cape Herschel ponds were excavated in granite and the loss of water was higher evap:precip ratios.  Although we published papers and reviews discussing loss of ecosystems with permafrost thaw, the Smol and Douglas 2007 PNAS paper is on ponds excavated in granite bedrock (so they are like bathtubs in the bedrock – not permafrost) – chosen as not directly influenced by permafrost thaw but increased evaporation. We had pondwater conductivity measures going back to 1983 – so we could show it was evaporation.  So, these ecosystems are disappearing due to evaporation, not permafrost thaw.

So it would be correct to say   by higher evaporation to precipitation ratios (Smol and Doulgas 2007) as well as permafrost thaw (many papers can be cited here…..  Chapter 7 of my new book Smol, J.P. 2023. *Lakes in the Anthropocene: Reflections on tracking ecosystem change in the Arctic.*  Excellence in Ecology Book Series, International Ecology Institute (ECI), Oldendorf/Luhe, Germany. 13 chapters. 438 pp. – has many examples.  One common one is:  Smith, L. C., Sheng, Y., MacDonald, G. M., and Hinzman, L.D. 2005. Disappearing Arctic lakes. Science 308: 1429,

*Response: As responded above, we completely agree and have revised the text accordingly, including also the suggested references plus some more.*

**Line 134**: add "a" before "tipping point"

**Line 142**: hysteresis should be plural "hystereses"

**Line 148**: remove "if not"; change "impacts" to "stressors"; remove "s" from "waters"

**Line 153**: change "among" to "as well as"

**Line 155**: change "pointed" to "identified"

**Line 160**: figure caption within the brackets: I found this hard to follow. Should "drive" here not be "driven"? "(...a tipping point-driven self-sustaining change)"?

Lines 165-169: this is an overly complex sentence that is hard to follow. Could you simplify/clarify?  (SEE ATTACHED PDF ---  MY CROSS-OUTS AND EDITS ARE NOT SHOWING IN PASTED WORD DOC)

Hysteresis can be strengthened by eutrophication-driven biological changes in biota, such as changes in fish composition and size structure with that have cascading effects on zooplankton and phytoplankton as well as strong impacts if on fish-mediated nutrient cycling (Brabrand et al. 1990). This in turn, also strengthen hysteresis and will maintain a system with deepwater anoxia and high nutrient load, supporting the release of GHGs (Fig. 2).

**Line 175:** Replace "speed" with "an acceleration"

**Line 176:** increases thermal stability and the duration and strength of stratification

**Line 177:** Minor, but there is a newer Woolway et al review that might be appropriate here: Woolway et al. 2022. Lakes in hot water: the impacts of a changing climate on aquatic ecosystems. BioScience 72: 1050-1061

**Line 187**: add a hyphen to "eutrophication-induced"

**Lines 186-190**: This is a very long and complex sentence – perhaps split in two.

**Line 201** (and elsewhere): Yang et al. (2015) is missing from the reference list.

**Lines 210-212**: Why only shallow lakes?

**Line 213:** what is meant by "coherent tipping"? Do you mean coherent "threshold exceedance"?

**Lines 215-218**: This sentence was long and complicated and I found it difficult to follow.

How about: (SEE ATTACHED PDF --- MY CROSS-OUTS AND EDITS ARE NOT SHOWING IN PASTED WORD DOC)

However, given the dearth of studies that generate bi-directional carbon flux data to assess the balance between emission and burial in lakes, it remains unknown whether the effect of any of eutrophication's climate feedback effect can be buffered by the projected eutrophication-driven increases in lake carbon burial (Anderson et al.2020). remains uncertain, and there is a dearth of studies that generate bi-directional carbon flux data to assess the balance between emission and burial in lakes.

**Line 221:** Consider starting a new sentence after (Grasset et al. 2020).

**Line 236**: There are several other studies relevant to treeline shifts and DOC or TOC including:

Pienitz, R. et al. 1999. Paleolimnological reconstruction of Holocene climatic trends from two boreal treeline lakes, Northwest Territories, Canada. *Arct, Antarct., and Alpine Res.* 31:82-93. https://doi.org/10.1080/15230430.1999.12003283

Rühland, K.M et al. Limnological characteristics of 56 lakes in the Central Canadian Arctic Treeline Region. J. Limnol. 62:9-27.

**Lines 247-250**: can you provide a reference for this. Also, the closed bracket is missing at end of sentence.

**Line 309**: delete "as of"

**Lines 310-312:** This sentence was difficult to understand as written. How about changing to:

(SEE ATTACHED PDF --- MY CROSS-OUTS AND EDITS ARE NOT SHOWING IN PASTED WORD DOC)

Given that high concentrations of DOM and deep-water anoxia are common, most Most boreal lakes are net heterotrophic and thus conduits of CO2, and often also CH4., due to high concentrations of DOM and common deep-water of sediment anoxia

**Lines 313-315**: This long sentence would be clearer if split into two. (SEE ATTACHED PDF --- MY CROSS-OUTS AND EDITS ARE NOT SHOWING IN PASTED WORD DOC)

If it eventually leads to oxygen depletion and cascading feedbacks then it would qualify as a tipping point. , yet with However, there would be a time delay between the two events, and where with the latter is being the critical tipping event.

**Line 327**: should this not be "drought" rather than "draught"?

**Line 329**: usually High Arctic is capitalized (but perhaps depends on the journal).

**Line 330**: delete "and" before "onset" and replace with "further promoting the onset of permafrost thaw..." (SEE ATTACHED PDF --- MY CROSS-OUTS AND EDITS ARE NOT SHOWING IN PASTED WORD DOC)

**Line 331**: delete "both" – the sentence starts with "Both"

**Line 339**: change "share" to "sheer"

**Line 344**: as noted above the Smol & Douglas paper is on evaporation – so I would change the sentence to: While the main problem is loss of water bodies affected by warming-induced increased evaporation rates (Smol and Douglas, 2007) and permafrost thaw (maybe cite Smith et al., 2005 here) ….

*Response:* Yes, done.

**Line 345-347**: In addition to "collapsing palsas and thermokarst areas" another climate-mediated phenomenon related to permafrost thaw is retrogressive thaw slumps that increase inorganic sediments to freshwater systems and affect biodiversity (Thienpont et al. 2013. Freshwater Biology; Heino et al. 2020).

**Lines 351-353**: The sudden introduction of "bird induced eutrophication" is a bit odd. Can you introduce/connect this better to the rest of this section – it seems to come out of nowhere. Also bird-induced here should be hyphenated.

**Line 361**: North America should not be hyphenated.

**Line 372**: delete extra period after reference.

**Line 461**: change "is" to "as"

**Line 497**: add an "s" to "regime shifts"

**Line 522 on**:  You might consider discussing "re-browning" here as well.

*Response:* This was considered, but for the flow of text we prefer to discuss re-browning early on, *L331-334.*

**Near lines 572 to 575**:  There is also the Smol et al 2005 PNAS compilation across the circum-polar Arctic that perhaps you meant to cite here?  It is in the reference list, but not actually cited in text as far as I can see… Also, the newer Kahlert et al. (2022) paper would be appropriate here: Kahlert, M. et al.  2022. Biodiversity patterns of Arctic diatom assemblages in lakes and streams: Current reference conditions and historical context for biomonitoring. Freshwater Biology 67: 116-140.

*Response: Yes, this is revised, including the inclusion of the Kahlert study: L645-648: Likewise regional patterns of species turnover (β-diversity) over 200 year demonstrated regional differences in species turnover, but also recent changes attributed to warming (Kahlert et al. 2020). As warming progresses, such studies serve as good references for future changes.*

---

## Author Comment (AC2)

**Response to Reviewers for Hessen et al., Lake ecosystem tipping points and climate feedbacks**

*We thank the reviewers for their positive judgment on our ms, and the very constructive comments. We outline below revisions in response to the general and specific points raised by each reviewer. We thank the reviewers for the suggestion to broaden the scope of the paper. We acknowledge that the original submission followed a definition of 'tipping point' applied in the report, 'Global Tipping Points' (Lenton et al., 2023) presented at the recent COP28-meeting (which is now cited); namely that there should be self-reinforcing feedbacks and evidence of hysteresis (as explained also in the ms). We have now explained this rationale in the manuscript along with considerations of broader contexts as recommended by the reviewers. A number of new references have been added to support our conclusions, and also to provide examples as suggested by RC1. The revisions are more than those pasted in the responses below, but the major revisions directly addressing suggestions or requests are refereed to with line number and revised text below.*

**Reviewer 1 (RC1)**

The contribution by Hessen et al summarises candidate tipping points (TP) in lake ecosystems focusing on their role and connection to climate feedbacks. The paper is something between a review/synthesis based (apparently) on the expert knowledge of the well-recognised author's list.

My main difficulty relates to the fact that it is not clear how the selection of the 6 candidate tipping points was made. The authors refer to the literate presenting tipping points but it is not clear what literature review was made, or whether these are already identified tipping points that the authors in this paper revisit according to their definition.

In this respect, it is feels a bit odd that out of the 6 candidate TP 3 are presented as not being TP (salinisation, shift to N-P limitation, invasive species).

Could these three make part of the discussion rather the main text?

*Response: We acknowledge that this could be confusing, and that the rationale for the selection both for the 6 candidate tipping points and the "chosen ones" should be clarified. This is now done in the revision, but in brief the author team, based on their wide experience in freshwater ecosystem research and insight in literature, proposed "candidates". We also searched the literature for hits on tipping points AND lakes (or ponds), and used this as an entry to the selection of relevant categories and cases. We agreed on 6 candidates with potentially widespread and relevant consequences. Among these, 2 could be considered as "real" tipping point categories according to the criteria of self-reinforcing feedbacks and hysteresis (rationale for definition now included in ms), 2 are type "binary shifts, and the last 2 represent potential threshold effects. We think all these cases are interesting by representing potentially widespread and to some extent climate driven non-linear responses, also often with repercussions to climate by GHG-emissions. This is further clarified in the revision.*

Line 59-69: Some types of changes can be classified as *binary*, i.e. either-or situations at the system level. Increased temperature and/or reduced precipitation may induce negative water balance and shrinking of water volumes to the level where lakes or ponds simply

disappear. Many lakes worldwide are facing reduced water volumes, but perhaps most striking is the widespread loss of high-latitude waterbodies, from Arctic or sub-Arctic ponds to wetlands or bogs. Such phenomena may qualify as one type of tipping point, but are not self-propelled by internal feedbacks *per se*, but rather by higher evaporation to precipitation ratios (Smol and Douglas 2007) or permafrost thaw (Smith et al. 2005; Webb et al. 2022; Smol 2023). While most tundra-ponds and other small waterbodies hardly qualify as *lakes* in the common sense, we mostly use the word lake through the text for simplicity, yet it will be evident from the context of wording where we specifically refer to ponds.

Now the discussion has an interesting part in connecting the first two TP (eutrophication, increased loading) but for the rest stays a bit generic on the issue of what is a tipping point or not for a lake in terms of persistence in time or abruptness.

*Response: We thank the reviewer for this important observation. This reflects that we arrived at two "real" tipping point categories according to the applied (and strict) criteria, but we feel that the discussion of the nature or type of non-linear responses is also important, since binary changes such as presence or absence of water bodies, salinization above tolerance thresholds for ecosystems (or key species), irreversible changes due to invasive species etc may pose mayor shifts in systems, partly caused by climate changes and partly with climate feedbacks. The text has been revised substantially and notably for the four non-TP categories. The changes are too many to be pasted below, but will appear from the revised ms per se.*

Also the waterbodies disappearance seems to be more of a tipping cascade rather a TP.

I miss a definition of binary shifts.

Also it is not clear what are tipping categories.

*Response: We agree, the disappearance of waterbodies is not presented as a TP in accordance with the strict definition. However, we do not consider this phenomenon to be 'cascade' per se, in that it is unlikely to be self-propelling. Binary shifts is an 'either-or' situation (presence-absence), and we have clarified this important point in the revised ms. See also the revised text above (L 59-69) as example. Also, in line with suggestions from ref. # 2, we have included negative water balance, and not only permafrost thaw, e.g. L 374-379-541: While the main problem is loss of water bodies affected by warming-induced increased evapotranspiration rates (Smol and Douglas 2007) and permafrost thaw (Smith et al. 2005), there are also cases where collapsing palsas and thermokarst areas create new waterbodies, and these waterbodies may themselves represent a positive feedback by accelerating the thaw (Langer et al. 2016; Turetsky et al., 2020).*

It would help to provide examples of actual lakes with the mentioned tipping points.

*Response: Many thanks. We acknowledge the space limitations but agree some real examples are important. We have now included real-lake examples for all categories, e.g. for eutrophication, L199-210: Warming also increases stratification and the duration and strength of stratification, also promoting anoxia (Maberly et al. 2020; Woolway et al. 2020; 2022). As a case example, this phenomenon is well documented by the recent study of the*

*Danish, shallow and highly eutrophic lake Ormstrup (Davidson et al. 2024). For browning (L 323-325): A well explored case study as an example of these impacts, which is also linked directly to the tipping point concept, is the Swedish, boreal brownwater lake Härsvatten where a long-term study clearly links loadings of DOC to anoxia (Spears et al. 2017). For loss of water-bodies (L380-388): Since most of these potentially lost waterbodies are small and nameless ponds, it is hard to point to specific cases, but the works cited above provide a number of telling examples. While the focus in this context is negative water balance or loss of high-latitude waterbodies, this is actually a widespread problem causing shrinking of many lakes. In Arctic areas, responses to warming may differ substantially between perennial lakes and ephemeral wetlands, related to ambient temperature and permafrost depth (Vulis et al. 2021). Although shrinking, appearance or loss of water bodies does not classify as a tipping event in the very strict sense, i.e. there is not obvious strong, self-reinforcing factors involved, it still is a climate driven result of climate change with potentially large, widespread and irreversible consequences.*

At the end of the increased DOM TP section, the authors conclude that there should be 2 TP considered: one to anoxia. How is this related to the TP eutrophication related anoxia? is it the same or different?

*Response: In principle it is the same, however the resulting internal P-loading is generally more prevalent in eutrophic lakes, but increased GHG-emissions will be the likely result in both cases. We have added clarification; L 344-349: If it eventually leads to oxygen depletion and cascading feedbacks then it would qualify as a tipping point. However, there would be a time delay between the two events, with the latter being the critical tipping event. The deep-water anoxia would in principle cause the same effects as in eutrophic lakes. the resulting internal P-loading is generally more prevalent in eutrophic lakes, but increased GHG-emissions will be the likely result in both cases.*

The feedbacks to the climate for the eutrophication and DOM TP could be made more clear.

I found the two paragraphs on it quite dense and hard to follow. It would be good to explicit describe the feedbacks between the climate and the lake ecosystems.

*Response: We have elaborated and clarified in the revised ms. Both in the introduction, the main text and the discussion, we now explicitly address why decrease in oxygen and increase in organic C and nutrients entail increased GHG-emissions*

Lastly, the text needs careful reading for syntax and typos.

*Response: We thank the reviewer for picking up on this. We have conducted a thorough check for syntax and typos.*